# A Structural Decomposition Analysis of China's Consumption-Based Greenhouse Gas Emissions

**Haidi Gao, Alun Gu, Gehua Wang and Fei Teng *** 

Institute of Energy, Environment and Economy (3E), Tsinghua University, Beijing 100084, China

\* Correspondence: tengfei@tsinghua.edu.cn; Tel.: +86-10-62784805

**Abstract:** The trends of consumption-based emissions in China have a major impact on global greenhouse gas (GHG) emissions. Previous studies have only focused on China's energy-related consumption-based emissions of $CO_2$ or specific non-$CO_2$ GHGs without taking overall consumption-based non-$CO_2$ GHG emissions into account. Based on a constructed global non-$CO_2$ GHG emissions database, combined with $CO_2$ emissions data, this paper fills this gap through an examination and analysis of China's GHG emissions using a global multi-regional input–output (MRIO) model for 2004, 2007 and 2011, and identifies the major factors driving changes in consumption-based emissions through a structural decomposition analysis (SDA). The results show that compared with $CO_2$ emissions, $CH_4$, $N_2O$ and F-gases emissions all increased more rapidly. Among consumption-based non-$CO_2$ GHG emissions, investment-based emissions experienced the fastest growth, but the net exports of non-$CO_2$ GHG emissions dropped drastically in recent years. While investment in total final consumption demand is the most influential factor for $CO_2$ emissions, household consumption most significantly affects the growth in consumption-based non-$CO_2$ GHG emissions.

**Keywords:** non-$CO_2$ greenhouse gases; consumption-based emissions; global multi-regional input–output model

## 1. Introduction

Global net anthropogenic $CO_2$ emissions must decline by about 45% from 2010 levels by 2030 and reach net zero around 2050, according to the October 2018 Intergovernmental Panel on Climate Change (IPCC) Special Report on the impacts of global warming of 1.5 °C above pre-industrial levels [1]. This requires the control of anthropogenic GHG emissions. Since China is the world's second largest economy and the largest GHG emitter, China's GHG emissions have an important impact on global GHG emissions.

In general, there are two accounting systems for measuring GHG emissions in a country or region: production-based emissions accounting and consumption-based emissions accounting [2]. Production-based emissions refer to emissions caused by all production activities within a country, including those emissions from domestic production for both domestic consumption and exports. Consumption-based emissions allocate emissions to all consumer activities within a country and encompass emissions from domestic consumption and emissions embodied in imports. By source, they can be divided into direct consumption-based emissions and indirect consumption-based emissions. Direct consumption-based emissions mean emissions from combustion activities and industrial production processes that use fossil energy such as coal, oil and natural gas. Indirect consumption-based emissions denote emissions other than direct consumption-based emissions from the perspective of the full life cycle, such as emissions associated with electricity and heat purchasing of residents in their daily lives. The gap between production-based and consumption-based emissions increases for most countries [3].

For the non-$CO_2$ GHG emissions that form an important part of total GHG emissions [4], inventories have also been developed at the global and national levels. Nevertheless, $CO_2$ emission abatement efforts far outweigh those for non-$CO_2$ GHG emissions. Non-$CO_2$ GHG emissions represent a smaller proportion of GHG emissions and fail to draw adequate attention in abatement policy formulation and modeling analysis. Additionally, non-$CO_2$ GHGs have a higher global warming potential (GWP) and can exist for hundreds of years in the atmosphere. To be consistent with existing inventory reporting guidelines, we use the GWPs from the 1996 reporting guidelines of the IPCC in this study. Relative to $CO_2$ emissions, there are opportunities to reduce non-$CO_2$ GHG emissions that would have a significant impact on overall GHG emission reductions.

This paper attempts to measure and analyze consumption-based GHGs in China. On the one hand, a database of non-$CO_2$ GHG emissions is built based on multiple sources of data, covering 140 countries and regions and 57 production sectors. It is combined with $CO_2$ emissions data from the Global Trade Analysis Project Version 9 (GTAP 9) database to form a global GHG emissions database. On the other hand, the indirect consumption-based emissions of various countries and regions are calculated using a global multi-regional input–output (MRIO) model. In summary, this study differs from existing literatures: first, the existing research only focuses on the accounting of consumption-based emissions for specific individual greenhouse gases ($CO_2$ or $CH_4$), while this study is the first to account for the consumption-based emissions of all greenhouse gases in China. Secondly, existing studies have not analyzed the main drivers behind changes in China's consumption-based emissions, while this paper uses the SDA method to analyze this in detail.

The following sections are organized as follows: Section 2 reviews the literature and Section 3 describes the research methods and data sources after the literature review in Section 2. Section 4 analyzes China's GHG emissions, consumption-based emissions, and the SDA results. Conclusions and policy recommendations are provided in the last section.

## 2. Literature Review

According to the EDGARv4.3.2 [5], China has been the world's largest production-based emitter of GHGs since 2004, so there are many studies specifically analyzing China's consumption-based emissions. Since 2015, several studies have estimated the differences and characteristics of consumption-based emissions from different industries, regions and different non-$CO_2$ GHGs in China and conducted input–output analyses (IOAs) based on China's national GHG emission inventories. Some studies calculated the country-specific year of peaking emissions by combining an IOA with climate policy modeling. Mi et al. [6] used a MRIO model to quantify the $CO_2$ emissions of 13 Chinese cities and analyze their production-based emissions, consumption-based emissions, and emissions per capita. This study pointed out that assuming an average gross domestic product (GDP) growth rate of over 5.0% over the next two decades, China will be able to peak its $CO_2$ emissions in 2026, considering the balance between $CO_2$ emissions reduction and economic trade growth. In the post-Paris era, urban emissions have received worldwide attention as cities are considered to be responsible for most of the world's environmental. Cities accommodate more than half of the world's population and are responsible for most of the global environmental footprints of carbon and resource consumption [7]. Based on the refined IOA approach, Mi et al. [8] assessed the potential impacts of industrial structures on energy consumption and $CO_2$ emissions through a case study on Beijing, and suggested that cap- and intensity-based targets should be balanced in emissions reduction strategies. Focusing on consumption-based emissions in different cities, Meng et al. [9] calculated and compared the production-based and consumption-based black carbon emissions in China's four municipalities (Beijing, Shanghai, Tianjin and Chongqing) and attributed the differences in emissions under the two accounting systems to differences in emissions intensity. Feng et al. [10] examined, from the perspective of consumption, the forces driving $CO_2$ emissions in the 30-year urbanization process of China's four municipalities. The results showed that consumption is associated with more than 70.0% of Beijing's $CO_2$ emissions and 48.0% of Chongqing's $CO_2$ emissions, whereas consumption-based emissions

occur outside city boundaries for Shanghai and Tianjin. Yang and Liu indicated that urban household carbon emissions were close to a 60/40 or 70/40 distribution; the economic features of different regions may contribute to such unequal distributions to a large extent [11]. With the continuous refinement of research, the measurement of household carbon emissions from a trading-oriented perspective has become increasingly important and urgently required due to the growing attention to personal carbon trading [12]. China is under great pressure to reduce $CO_2$ emissions and China's economy is more reliant on domestic consumption (particularly household consumption) than on exports. So, there is an urgent need to study $CO_2$ emissions from household consumption [13]. Shao et al [14] revealed the carbon imbalances of most Chinese provinces and cities decreased between 2007 and 2010, but disparities in the regional per capita carbon footprint widened. Indirect carbon emissions from households can be redefined and recalculated using an input–output model [15].

Given refined statistics and studies on production-based non-$CO_2$ GHG emissions [4], more researchers have turned their attention to consumption-based non-$CO_2$ GHG emissions. According to official 2007 statistics, China's $CH_4$ emissions were estimated to total 39.6 $MtCO_2e$, which is equivalent to three quarters of China's $CO_2$ emissions from fuel combustion by the global thermodynamic potential [16]. According to the 1990–2010 F-gases inventory data, the electrical equipment sector contributed most (about 70.0%) to $SF_6$ emissions, followed by magnesium production, semiconductor manufacturing, and other emitting sectors (about 10.0% each). With a GWP 23,900 times that of $CO_2$, $SF_6$ produces a greater greenhouse effect compared to other GHG emissions such as $CO_2$, $CH_4$ and $N_2O$ [17]. There are also studies that measured the GHG emissions of provinces and sectors in China while taking the national emissions flows into account. Among these studies, Liu et al. [18] investigated the differences and characteristics of Chinese regions and sectors in GHG emissions trends, and concluded that the regional disparity in technology has become a major obstacle to China's GHG emission reduction.

For further examining the forces driving indirect consumption-based GHG emissions, SDA is a prevalent tool. In general, both SDA and index decomposition analysis (IDA) are applicable to such studies. By virtue of its simple methodology and low data threshold, IDA is widely used in the analysis of energy demand, industrial structures, and the influencing factors of $CO_2$ emissions [19], but the method cannot further decompose the production technology and final demand structure for input–output analysis [20]. SDA can make up for IDA's deficiencies in these aspects, as detailed by many comparative studies on IDA and SDA [21,22]. It can be combined with an IOA to investigate the impacts on indirect consumption-based GHG emissions brought by changes in many factors, including production technology, emissions intensity, and final consumption demand structure. Therefore, this paper uses SDA to calculate and analyze indirect consumption-based GHG emissions. There are a wide range of SDA applications for analysis of China's GHG emissions at the national and regional levels. Among representative national-level SDA studies, Su et al. [23] examined China's GHG emissions at different levels across regions and time, and he used the IOA approach to analyze the aggregate embodied intensity (AEI) of China's $CO_2$ emissions in 2012 from the perspective of consumption, and identified the driving forces for these emissions through multiplying the SDA analysis of AEI changes at different levels. Su et al. [24] divided China into eight regions for comparative SDA analysis and found that the results were highly dependent on spatial aggregation. Furthermore, Su et al. [25] performed a comparative SDA study of 30 regions in China, and ranked them according to the chess room effect. Su et al. [26] also built a 2006–2012 input–output database for input–output analysis, and by means of SDA revealed the factors driving China's trade changes during the period. In addition, Chen [27], Fan [28], Fang et al. [29], and Guo [30] discussed China's $CO_2$ emissions; Bian [31], Fu and Li [32], Qu et al. [33], and Zhu et al. [34] probed into the indirect consumption-based emissions associated with Chinese residents. Some studies have focused on $CO_2$ emissions embodied in China's trade [35–57]. Among representative regional-level SDA studies, Wei et al. [58] analyzed the input–output relationship of energy-related $CO_2$ emissions from Beijing's industrial sector from 2000–2010, and evaluated the impacts on $CO_2$ emissions by technology, sector,

economic structure, and economies of scale. In addition, China tends to shift gradually from an investment- to a consumption-driven economy [59].

Given there are few studies on consumption-based non-$CO_2$ GHG emissions, this paper examines China's consumption-based GHG emissions and their influencing factors, with a focus on the composition of these emissions. It encompasses a comparative analysis of production-based emissions, consumption-based emissions, and consumption-based emission intensity, and a quantitative analysis of China's indirect consumption-based emissions based on SDA theory.

## 3. Research Methods and Data Sources

### 3.1. Research Methods

#### 3.1.1. Global MRIO Model

In line with the Peters' et al. methodology [60], a global MRIO model is proposed for analyzing consumption-based emissions, and the MRIO table is drawn based on data from non-competitive input–output tables of various countries and regions in the GTAP 9 database. The table takes the global economy as a whole and covers the economic structure and trade of 140 countries and regions around the world, involving 57 production sectors. The latest available GTAP database is the GTAP 9 published in year 2015, and the base years are 2004, 2007 and 2011.

According to the input–output relationship, the horizontal balance of the global MRIO model can be expressed by Equation (1).

$$X_i = AX_i + Y_i \tag{1}$$

Equation (1) indicates the non-competitive input–output balance of a country or region $i$. $X_i$ represents the magnitude of value of total output in a country or region $i$. $A$ is the intermediate matrix of trade between production sectors in a country or region $i$, including the coefficient matrix of domestic intermediate consumption in country or region $i$, and the coefficient matrix of intermediate consumption of exports to other countries and regions. $Y_i$ represents the vector of total final consumption demand in a country or region $i$, covering the final demand for domestic consumption in various sectors of a country or region $i$ and for imports from other countries and regions. It can be further broken down into the column vector of government consumption $Y_{iG}$, column vector of investment $Y_{iI}$, and column vector of household consumption $Y_{iP}$, so $Y = Y_{iG} + Y_{iI} + Y_{iP}$.

$$X_i = (I - A)^{-1} Y_i \tag{2}$$

Equation (2) is a variant of Equation (1). $I$ is the unit matrix, also referred to as the Leontief inverse matrix $L = (I - A)^{-1}$. The matrix can be used to calculate the products and services that transform into final demand. It can track the demand of various sectors in the production process and the intermediate demand for products and services to calculate comprehensive statistics on all consumer demands.

Under the standard IOA framework for energy and the environment, the economic output is quantified, and then the emissions and impacts of economic activities are calculated based on the emissions factors. For GHG emissions, the relevant emission factor is the GHG emission per unit of economic output, that is, the ratio of total emissions to total output of production sectors in a country or region. Therefore, the following equation can be obtained.

$$E_i = F(I - A)^{-1} Y_i \tag{3}$$

Equation (3) indicates the indirect consumption-based emissions from all production in a country or region $i$. Represented by $Y_i$, these emissions include indirect emissions from domestic consumption in a country or region $i$ and indirect emissions from consumption of exports from other countries and regions. Indirect consumption-based emissions are calculated by the sector-specific GHG emissions

intensity vector $F$ in a country or region $i$. The calculations redistribute the indirect consumption-based emissions of global products according to the direction and volume of consumption-based trade flows, and when combined with the direct consumption-based emissions, indicate the consumption-based emissions of various countries and regions. In the following research, GHG emissions are divided into $CO_2$ emissions and non-$CO_2$ GHG emissions based on the emissions factors of different gases.

### 3.1.2. Global MRIO-SDA method

Equation (4) represents the SDA analysis of four influencing factors in the global MRIO model. The subscripts 0 and 1 denote two time points in temporal SDA and two different countries or regions in spatial SDA.

$$\Delta E = F_1 L_1 Y_1 - F_0 L_0 Y_0 \tag{4}$$

The final consumption demand $Y$ is further decomposed in a similar way with the emissions intensity $F$ of GHG emissions. From $Y$, the total final consumption demand of 140 countries and regions can be calculated based on the final consumption of various sectors, expressed by scalar $y_t$. The structure of final consumption demand of sectors in countries and regions is expressed by vector $y_s$. The sum of this vector of 140 countries and regions equals 1, as shown in Equation (5).

$$Y = y_t * y_s \tag{5}$$

Therefore, combined with GHG emissions intensity $F$ and final consumption demand $Y$, the global MRIO model can be decomposed through SDA into four influencing factors, as shown in Equation (6).

$$E = F * L * y_t * y_s \tag{6}$$

In order to reduce the error caused by crossing terms, the two-pole decomposition method is applied to obtain Equation (7).

$$\Delta E = \frac{1}{2}\Delta F * L_1 * y_{t,1} * y_{s,1} + \frac{1}{2}\Delta F * L_0 * y_{t,0} * y_{s,0}$$

$$+ \frac{1}{2}F_0 * \Delta L * y_{t,1} * y_{s,1} + \frac{1}{2}F_1 * \Delta L * y_{t,0} * y_{s,0}$$

$$+ \frac{1}{2}F_0 * L_0 * \Delta y_t * y_{s,1} + \frac{1}{2}F_1 * L_1 * \Delta y_t * y_{s,0}$$

$$+ \frac{1}{2}F_0 * L_0 * y_{t,0} * \Delta y_s + \frac{1}{2}F_1 * L_1 * y_{t,1} * \Delta y_s \tag{7}$$

Therefore, the global MRIO-SDA analysis of four influencing factors can be written as Equation (8).

$$\Delta E' = \Delta F' + \Delta L' + \Delta y_t' + \Delta y_s' \tag{8}$$

It shows the four specific factors affecting global indirect consumption-based GHG emissions. Specifically, $\Delta F'$ represents the effect of GHG emissions intensity (per unit of output), $\Delta L'$ the effect of intermediate production technology, $\Delta y_t'$ the effect of global final consumption demand, and $\Delta y_s'$ the effect of final consumption structure of countries and regions. Among these, $\Delta F'$ and $\Delta L'$ combined indicate the production-based effect, while $\Delta y_t'$ and $\Delta y_s'$ combined form the consumption-based effect. In short, the SDA analysis identifies four factors influencing indirect consumption-based emissions. Each factor can be divided into a government, investment, and household contribution according to GHG emissions so as to accurately locate the sources of influencing factors. Usually, the household is an important sector for resource consumption and environmental impact [61].

*3.2. Data Sources*

The data used in this paper encompasses economic data and emissions data. The emissions data are sourced from the GTAP 9 $CO_2$ emissions database calibrated to the International Energy Agency (IEA) data. Non-$CO_2$ GHG emissions are from the database built in this paper. The data sources of non-$CO_2$ GHGs are from the Second National Communication on Climate Change of the People's Republic of China [62] and the People's Republic of China First Biennial Update Report on Climate Change [63]. According to relevant decisions of United Nations Framework Convention on Climate Change (UNFCCC), and considering China's circumstances, the National Greenhouse Gas Inventory of 2005 and 2012 covers six gases including carbon dioxide ($CO_2$), methane ($CH_4$), nitrous oxide ($N_2O$), hydrofluorocarbons (HFCs), perfluorocarbons (PFCs) and sulfur hexafluoride ($SF_6$) from energy, industrial processes, agriculture, land use change and forestry and waste. The Inventory mainly follows the Revised 1996 IPCC Guidelines for National Greenhouse Gas Inventories (hereinafter referred to as the Revised 1996 IPCC Guidelines) and the IPCC Good Practice Guidance and Uncertainty Management in National Greenhouse Gas Inventories (hereinafter referred to as the IPCC Good Practice Guidance). Activity data are mainly from official statistics, while emission factors are mainly from the 2012 China's country specific parameters [63]. In addition, this paper uses the IPCC definition code for non-$CO_2$ GHG activities and China's GDP growth rate of World Bank Open Data to adjust non-$CO_2$ GHG activities and emissions to 2004, 2007 and 2011 reference years, as well as 57 GTAP sectors. The economic data are derived from the global non-competitive input–output tables for 140 countries and regions and 57 sectors covered by GTAP 9 database. This also includes relevant data about intermediate production technology in the input–output model and about total final consumption demand (specifically including government consumption, household consumption, and investment). Since the SDA time steps are consistent with the base years of the global MRIO model, i.e., 2004, 2007, and 2011, the SDA analysis examines consumption-based indirect GHG emissions during the periods from 2004 to 2007 and from 2007 to 2011.

The accurate calculation of consumption-based emissions must be based on an accurate inventory of current emissions and input–output data. However, the accuracy of non-$CO_2$ inventory is much less certain than that of $CO_2$ inventory. For example, the accuracy of China's non-$CO_2$ inventories ranges from ±55%–60% at the high end to nearly ±15% at the low end, compared with an uncertainty range of ±6% for China's $CO_2$ emissions. Due to the lack of statistical data, it is currently impossible to quantify the uncertainty range of accounting results of consumption-based emissions. Improving the accuracy of inventory data can greatly help improve the accuracy of consumption-based emissions accounting

## 4. Result and Discussion

*4.1. The Realted Analysis of GHG Emissions in China*

From 2004 to 2011, both production-based and consumption-based GHG emissions in China were on the rise, with increments of 4.2 $GtCO_2e$ and 4.0 $GtCO_2e$ (from 4.7 $GtCO_2e$ to 8.7 $GtCO_2e$), respectively, and with an increase of 67.8% and 83.6%, respectively. Thus, there was a greater increment of production-based emissions, but a higher growth rate of consumption-based emissions. During this period, China's consumption-based GHG emissions grew, which is mainly attributed to $CO_2$ emissions. China's consumption-based GHG emissions expanded continuously by 83.6% during 2004–2011, and represented 13.4%, 15.9% and 20.9% of the global GHG emissions total in 2004, 2007, and 2011 respectively. Specifically, $CO_2$ emissions increased from 3.6 $GtCO_2e$ to 7.0 $GtCO_2e$, $CH_4$ from 0.75 $GtCO_2e$ to 1.0 $GtCO_2e$, $N_2O$ from 0.3$GtCO_2e$ to 0.5 $GtCO_2e$, and F-gases from 0.07 $GtCO_2e$ to 0.2 $GtCO_2e$, an increase of 94.6%, 39.3%, 57.0% and 114.0% over this time period. From the perspective of consumption-based emission increments, $CO_2$ emissions, with an additional 3.4 Gt, contributed to 87.5% of incremental consumption-based GHG emissions in China during 2004–2011.

During 2004–2011, China's net exports of non-$CO_2$ GHG emissions dropped drastically. As shown in Figure 1, China's GHG emissions are divided into direct consumption-based emissions, indirect

consumption-based emissions, and net exports of emissions. For both $CO_2$ and non-$CO_2$ GHGs, both direct and indirect consumption-based emissions gradually climbed during 2004–2011, while the net exports of emissions declined after an initial growth spurt. On the whole, consumption-based GHG emissions stayed much lower than production-based GHG emissions, although both increased in China. The net exports of GHG emissions increased first and then decreased, and the net exports of non-$CO_2$ GHG emissions fell sharply by 55.2% in 2011 from the 2007 level. Meanwhile, in 2011, for example, the emissions related to government consumption, household consumption, and investment numbered 0.8 $GtCO_2$e, 4.1 $GtCO_2$e and 3.9 $GtCO_2$e, respectively, but the direct emissions based on government and household consumption were 2378.0 $tCO_2$e and 0.6 $GtCO_2$e only. This implies that indirect emissions are a major part of consumption-based emissions.

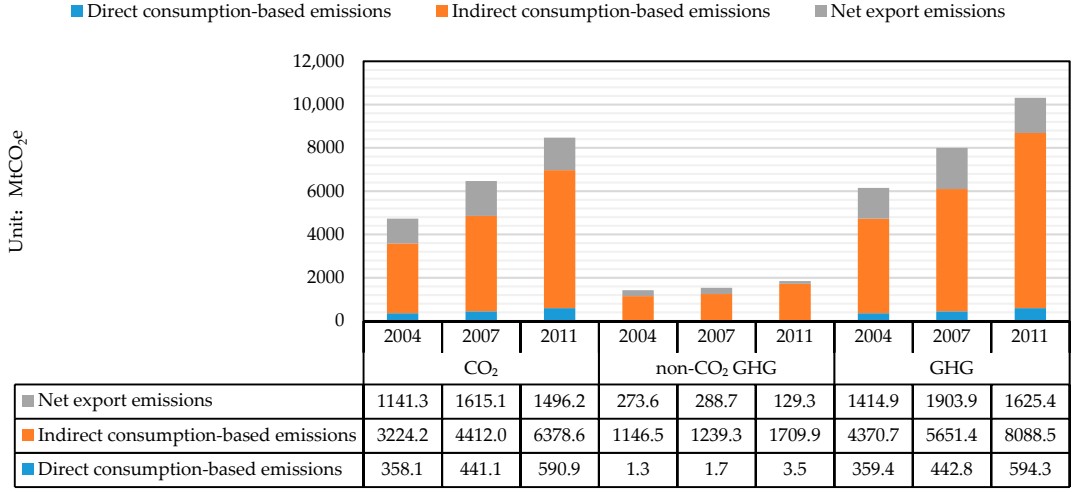

| | 2004 | 2007 | 2011 | 2004 | 2007 | 2011 | 2004 | 2007 | 2011 |
|---|---|---|---|---|---|---|---|---|---|
| | | $CO_2$ | | | non-$CO_2$ GHG | | | GHG | |
| ■ Net export emissions | 1141.3 | 1615.1 | 1496.2 | 273.6 | 288.7 | 129.3 | 1414.9 | 1903.9 | 1625.4 |
| ■ Indirect consumption-based emissions | 3224.2 | 4412.0 | 6378.6 | 1146.5 | 1239.3 | 1709.9 | 4370.7 | 5651.4 | 8088.5 |
| ■ Direct consumption-based emissions | 358.1 | 441.1 | 590.9 | 1.3 | 1.7 | 3.5 | 359.4 | 442.8 | 594.3 |

**Figure 1.** China's production-based and consumption-based emissions and consumption-based direct and indirect emissions.

The percentage change is analyzed in depth for China's production-based and consumption-based GHG emissions. Table 1 gives the relevant data. The difference between production-based emissions and consumption-based emissions is: although both production-based and consumption-based $CO_2$ emissions grew continuously, growth slowed down for the former, but picked up for the latter. As for non-$CO_2$ GHG emissions, growth quickened from both the production and consumption perspectives. The percentage change for different emissions (either production-based or consumption-based) during the same period is compared. For $CO_2$ emissions, the percentage change for each period declined from 36.9% to 35.5% from 2004–2007, but increased from 30.9% to 43.6% from 2007–2011. For non-$CO_2$ GHG emissions, the percentage change was on the rise in the two periods, up from 7.6% to 8.1% and from 20.5% to 38.1% (an increase of 17.6%), respectively. Meanwhile, the percentage change in the same emissions during different periods is compared. The percentage change in production-based $CO_2$ emissions slowed down from 36.9% to 30.9%, while that of consumption-based $CO_2$ emissions intensified from 35.5% to 43.6%. For non-$CO_2$ GHGs, the percentage change in production-based emissions soared from 7.6% to 20.5% and consumption-based emissions from 8.1% to 38.1% (an increase of up to 30.0%). As $CO_2$ takes up a relatively large proportion of the GHG emissions, the percentage change in GHG emissions tends to be close to that of $CO_2$ emissions.

**Table 1.** China's GHG emissions in 2004, 2007 and 2011.

| MtCO$_2$e | | Production-Based Emissions | Consumption-Based Emissions | Direct Household | Indirect Household | Investment | Direct Government | Indirect Government |
|---|---|---|---|---|---|---|---|---|
| 2004 | CO$_2$ | 4723.6 | 3582.3 | 358.1 | 1310.4 | 1606.1 | 0.0 | 307.6 |
| | CH$_4$ | 905.5 | 749.5 | 0.3 | 440.3 | 216.8 | 0.0 | 92.3 |
| | N$_2$O | 385.6 | 322.6 | 1.0 | 211.7 | 74.8 | 0.0 | 36.2 |
| | F-gases | 130.2 | 74.4 | 0.0 | 28.6 | 35.1 | 0.0 | 10.7 |
| | GHG | 6144.9 | 4728.8 | 359.4 | 1991.1 | 1932.8 | 0.0 | 446.8 |
| 2007 | CO$_2$ | 6468.3 | 4853.1 | 441.1 | 1790.5 | 2196.5 | 0.0 | 425.0 |
| | CH$_4$ | 954.9 | 803.5 | 0.5 | 447.1 | 247.4 | 0.0 | 109.0 |
| | N$_2$O | 387.6 | 335.5 | 1.2 | 215.0 | 81.9 | 0.0 | 38.5 |
| | F-gases | 187.3 | 100.3 | 0.0 | 35.9 | 51.0 | 0.0 | 13.4 |
| | GHG | 7998.0 | 6092.5 | 442.8 | 2488.6 | 2576.9 | 0.0 | 585.9 |
| 2011 | CO$_2$ | 8465.6 | 6969.5 | 590.9 | 2364.7 | 3466.4 | 0.0 | 547.5 |
| | CH$_4$ | 1112.0 | 1044.2 | 1.9 | 530.5 | 375.9 | 0.0 | 137.8 |
| | N$_2$O | 558.3 | 506.5 | 1.5 | 314.5 | 141.8 | 0.0 | 50.2 |
| | F-gases | 172.3 | 159.2 | 0.0 | 52.0 | 88.3 | 0.0 | 18.9 |
| | GHG | 10,308.3 | 8679.4 | 594.3 | 3261.7 | 4072.3 | 0.0 | 754.5 |

From the consumption perspective, China reached the global level of per capita GHG emissions later than when accounting for the production perspective. China's position in the global rankings by per capita GHG emissions is far from its position by total GHG emissions. According to IEA statistics, in terms of GHG emissions, China ranked first among countries and regions around the world by producing 6.0 GtCO$_2$e emissions in 2006. Global annual per capita emissions in 2004, 2007 and 2011 were 5.5 tCO$_2$e, 5.8 tCO$_2$e and 6.0 tCO$_2$e, respectively. From a production point of view, China's per capita emissions, at 6.1 tCO$_2$e, exceeded the global average in 2007. From a consumption point of view, however, exceeding the average did not occur until 2011, when China's per capita emissions reached 6.5 tCO$_2$e per year. China's GHG emissions grew quickly mainly in the fields of energy, transportation, and trade.

### 4.2. Composition of China's Consumption-Based GHG Emissions

Consumption-based emissions can be divided into direct consumption-based emissions and indirect consumption-based emissions, and they are attributed to household consumption, investment, and government consumption according to sectors of emissions. In 2011, for example, direct emissions formed a small part of consumption-based GHG emissions, while indirect emissions occupied a large proportion. Specifically, the direct emissions relating to government consumption and household consumption accounted for 0.0% and 9.8%, respectively. The indirect emissions related to government consumption, investment, and household consumption, which stood at 4.2 GtCO$_2$e, 10.0 GtCO$_2$e, and 23.4 GtCO$_2$e, represented 10.1%, 23.9% and 56.2% of Chinese total consumption-based emissions, respectively. Among these, the largest was household consumption-based emissions.

Investment-based emissions experienced the largest growth in China's consumption-based CO$_2$ emissions. As shown in Figure 2, China's consumption-based CO$_2$ emissions totaled 3.6 Gt, 4.9 Gt, and 7.0 Gt in 2004, 2007 and 2011, respectively, and rose by 35.5% and 43.6% during the two in-between periods. According to sectors of emissions, they included emissions related to government consumption, emissions related to household consumption, and investment-based emissions, wherein the former two can be further divided into direct and indirect emissions. From 2004 to 2007, the three components expanded by 28.2%, 33.7% and 36.8%, respectively, and from 2007 to 2011, the increases were 28.8%, 32.4% and 57.8%. By comparison, investment-based emissions had the most significant growth, which was larger than that of consumption-based CO$_2$ emissions during both periods.

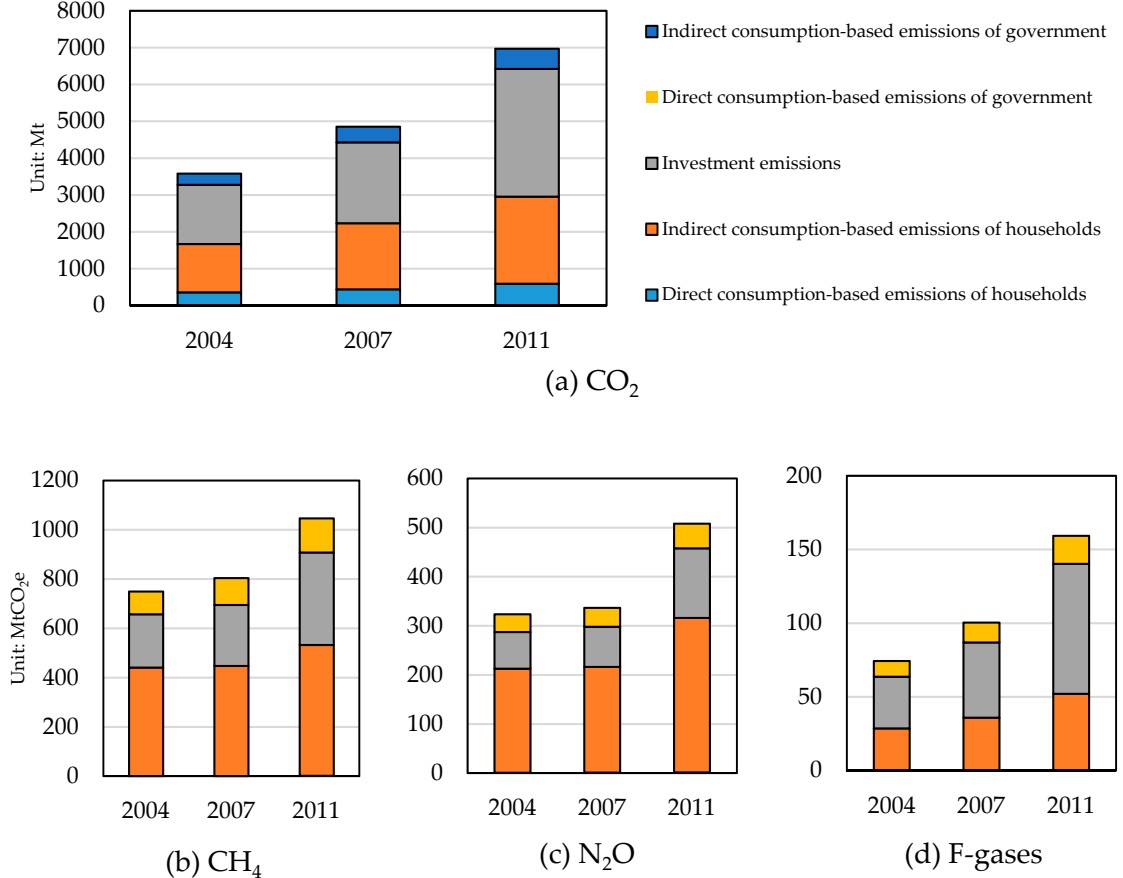

**Figure 2.** Composition of China's consumption-based (**a**) $CO_2$, (**b**) $CH_4$, (**c**) $N_2O$, and (**d**) F-gases in 2004, 2007, and 2011.

Investment-based emissions and household consumption-based emissions exhibited the largest and fastest growth in China's consumption-based $CH_4$ emissions. China's consumption-based $CH_4$ emissions amounted to 0.75 $GtCO_2e$, 0.8 $GtCO_2e$ and 1.0 $GtCO_2e$ in 2004, 2007 and 2011 respectively. They were up by 7.2% and 30.1% during the two in-between periods, respectively, and soared by 316.4% across the whole period. Compared with the study by Zhang et al [64], China's $CH_4$ consumption-based emissions in 2012 using the EDGARv4.3.2 database were 0.7 $GtCO_2e$, less than 0.3 $GtCO_2e$ emissions calculated from this paper. This is largely due to the difference between the EDGAR database and the Second National Communication on Climate Change of the People's Republic of China and the People's Republic of China First Biennial Update Report on Climate Change. Specific to sectors of emissions, growth registered 18.1%, 14.1% and 1.6% during 2004–2007 for emissions related to government consumption, investment, and household consumption respectively, and grew to 26.4%, 51.9% and 19.0% during 2007–2011. By comparison, from 2004–2011, investment-based emissions experienced the greatest growth at 37.8%, followed by emissions related to household consumption and government consumption. During 2007–2011, household consumption-based emissions accelerated relative to the previous period. The between-period growth rate increased most rapidly, from 1.6% to 19%, followed by investment-based emissions and government consumption-based emissions. Investment boosts economic development by stimulating consumer demand and expanding international trade. As the world's largest manufacturer and exporter, China's investment-based $CH_4$ emissions expanded at a much larger magnitude than consumption-based $CH_4$ emissions as a whole. This means the massive production and consumption associated with investment not only meets the needs of domestic consumption, but also adapts to changing international trade. At the same time, consumption drives production. The rapid increase of household consumption-based emissions denotes growth in the final

consumption demand of households, and reflects the rapid growth of domestic final consumption demand in China.

Investment-based emissions are also the fastest-growing part of China's consumption-based $N_2O$ emissions. China's consumption-based $N_2O$ emissions added up to 0.32 $GtCO_2e$, 0.34 $GtCO_2e$, and 0.5 $GtCO_2e$ in 2004, 2007 and 2011, respectively. They rose by 4.0% during 2004–2007 and 50.9% during 2007–2011. To break it down, emissions related to government consumption, investment, and household consumption increased by 6.4%, 9.6% and 1.7% during the first period, respectively, and 30.4%, 73.1% and 46.1% during the second period. Among these, investment-based emissions showed the greatest growth, which outweighed that of total consumption-based $N_2O$ emissions during both periods. Emissions related to household and government consumption also became significantly larger during 2007–2011.

Meanwhile, investment-based emissions experienced the fastest growth in China's consumption-based F-gases emissions. China's consumption-based F-gases emissions numbered 0.07 $GtCO_2e$, 0.1 $GtCO_2e$, and 0.2 $GtCO_2e$ in 2004, 2007 and 2011 respectively, and increased by 34.9% and 58.7% during the two in-between periods respectively. For emissions related to government consumption, investment and household consumption, the growth registered 25.8%, 45.4% and 25.5% during 2004–2007, and 41.1%, 73.0% and 44.9% during 2007–2011. Investment-based emissions expanded most markedly among these three components, and outpaced consumption-based F-gases emissions as a whole during both periods. Emissions related to household and government consumption also grew during 2007–2011.

*4.3. SDA of China's Indirect Consumption-Based GHG Emissions*

Investment demand is the most influential among the factors driving China's $CO_2$ emissions. There are four factors that affect China's indirect consumption-based $CO_2$ emissions: emissions intensity, intermediate production technology, total final consumption demand, and final consumption demand structure. This paper examines the impact direction and magnitude of the four factors on emissions during the periods from 2004 to 2007 and from 2007 to 2011. As shown in Figure 3, total final consumption demand had a positive effect on $CO_2$ emissions, including investment, household consumption, and government consumption, in descending order of effect. Among them, investment contributed to 97.0% and 129.3% of indirect consumption-based $CO_2$ emissions during the two periods, respectively. Both emissions intensity and intermediate production technology exerted negative effects, especially investment-related components. With the improvement of production efficiency and the development of more low-carbon production technologies and their equipment, the two factors lowered $CO_2$ emissions in production processes. Therefore, the SDA of China's indirect consumption-based $CO_2$ emissions proves from another research perspective that China's $CO_2$ emissions are mainly associated with investment. This is related to China's economic policy of focusing on developing infrastructure and improving residents' living standards during this period.

Corresponding to $CO_2$ emissions, non-$CO_2$ GHG emissions are divided into $CH_4$, $N_2O$ and F-gases in accordance with gas types and sectors covered by the database constructed in this paper. Therefore, the causes of specific non-$CO_2$ GHG emissions and the factors influencing indirect consumption-based emissions are analyzed by sector.

Household final consumption demand is the most important factor affecting China's $CH_4$ emissions. From 2004 to 2007, total final consumption demand, particularly household consumption, exerted a significant positive effect on the emissions, while emissions intensity and intermediate production technology influenced the emissions in the other direction; noticeably so in the emissions intensity of the household sector. During this period, China's $CH_4$ emissions grew by 54.1 $MtCO_2e$ overall. Changing emissions intensity reduced $CH_4$ emissions by 358.3 $MtCO_2e$, or 209.4 $MtCO_2e$, 44.1 $MtCO_2e$, and 104.8 $MtCO_2e$ in the household, government, and investment sectors, respectively. The percentage change in emissions due to emissions intensity reached −662.7% overall, or −387.3%, −81.5%, and −193.9% from each sector, respectively. Intermediate production technology helped

cut $CH_4$ emissions by 32.6 $MtCO_2e$, which equaled a −60.3% change in emissions. Within this amount, the reductions in the household, government, and investment sectors numbered 24.6 $MtCO_2e$, 1.7 $MtCO_2e$ and 6.3 $MtCO_2e$ and represented a −45.4%, −3.2% and −11.7% reduction, respectively. Total final consumption demand drove $CH_4$ emissions up by 448.1 $MtCO_2e$, of which 242.0 $MtCO_2e$, 62.4 $MtCO_2e$, and 141.7 $MtCO_2e$ occurred in the household, government, and investment sectors, respectively. The percentage of change in emissions was 828.9% overall or 447.6%, 115.4%, and 265.9% from each sector, respectively. The final consumption demand structure contributed to −2.9% of the $CH_4$ emissions by reducing 3.2 $MtCO_2e$. Specifically, emissions related to household consumption and investments were reduced by 1.3 $MtCO_2e$ and 2.0 $MtCO_2e$, and emissions related to government consumption added 0.07 $MtCO_2e$, so the percentages of change in emissions were −2.4%, −3.7% and 0.1%, respectively. From 2007 to 2011, China's $CH_4$ emissions were increased by the intermediate production technology and total final consumption demand, but were reduced by the emissions intensity, especially in the household sector. During this period, China's $CH_4$ emissions increased by 240.7 $MtCO_2e$ overall. Affected by the emissions intensity, $CH_4$ emissions were reduced by 448.9 $MtCO_2e$, including 254.2 $MtCO_2e$, 61.7 $MtCO_2e$, and 133.0 $MtCO_2e$ in the household, government, and investment sectors, respectively. They contributed −186.5%, −105.9%, −25.6% and −55.3% of the overall change in $CH_4$ emissions, respectively. Intermediate production technology raised the $CH_4$ emissions by 15.3 $MtCO_2e$, contributing to 6.3% of the overall change in $CH_4$ emissions. For this factor, the increases in the household, government, and investment sectors numbered 11.2 $MtCO_2e$, 0.6 $MtCO_2e$, and 3.5 $MtCO_2e$ and represented 4.7%, 0.2% and 1.4% of the overall change in $CH_4$ emissions, respectively. For the total final consumption demand, $CH_4$ emissions grew by 725.5 $MtCO_2e$, wherein 360.6 $MtCO_2e$, 89.7 $MtCO_2e$, and 275.2 $MtCO_2e$ were associated with the household, government, and investment sectors, respectively. The respective percentages of change in $CH_4$ emissions registered 304.1% total, with 149.8%, 39.3%, and 114.3% from each sector. The final consumption demand structure led to a decline of 51.2 $MtCO_2e$ in the $CH_4$ emissions, accounting for −21.3% of the overall change in $CH_4$ emissions. Among these, emissions related to household consumption and investment fell by 34.1 $MtCO_2e$ and 17.3 $MtCO_2e$ respectively and emissions related to government consumption increased by 0.2 $MtCO_2e$, which represented −14.2%, −7.2% and 0.1% of the overall changes in $CH_4$ emissions, respectively.

Household final consumption demand, among all factors, most affects China's $N_2O$ emissions. From 2004 to 2007, the total final consumption demand, especially household consumption, exerted an upward effect on China's $N_2O$ emissions, while the most influential downward factors were the emissions intensity and intermediate production technology, especially for the household sector. During this period, China's $N_2O$ emissions grew by 12.8 $MtCO_2e$ overall. For the emissions intensity, $N_2O$ emissions were decreased by 143.8 $MtCO_2e$, with 88.5 $MtCO_2e$, 19.5 $MtCO_2e$, and 35.7 $MtCO_2e$ of the decrease in the household, government, and investment sectors, respectively. The respective percentages of change of the $N_2O$ emissions over this time period were −1121.4% overall, and −690.5%, −152.2%, and −278.7% for each sector respectively. Intermediate production technology reduced $N_2O$ emissions by 21.1 $MtCO_2e$, contributing to −164.7% of the overall change in the $N_2O$ emissions. Within this, the reductions in the household, government, and investment sectors stood at 15.0 $MtCO_2e$, 1.7 $MtCO_2e$, and 4.4 $MtCO_2e$, and represented −116.7%, −13.3% and −34.7% of the overall change in the $N_2O$ emissions respectively. Total final consumption demand drove the $N_2O$ emissions up by 188.1 $MtCO_2e$, wherein 115.8 $MtCO_2e$, 23.5 $MtCO_2e$, and 48.8 $MtCO_2e$ were attributed to household consumption, government consumption, and investment, respectively. The respective percentages of change in $N_2O$ emissions were 304.1% total or 149.8%, 39.3% and 114.3% in each sector. Due to final consumption demand structure, the $CH_4$ emissions decreased by 10.3 $MtCO_2e$, of which 8.9 $MtCO_2e$ and 1.5 $MtCO_2e$ were related to household consumption and investment, despite an increase of 0.02 $MtCO_2e$ related to government consumption. The percentages of change in $N_2O$ emissions reached −80.6%, −69.4%, −11.3% and 0.2%, respectively. From 2007 to 2011, China's $N_2O$ emissions, especially in the household sector, increased by the total final consumption demand, but were decreased

by the emissions intensity and intermediate production technology. During this period, China's $N_2O$ emissions increased by 171.0 $MtCO_2e$. Due to emissions intensity, the $CH_4$ emissions decreased by 95.0 $MtCO_2e$, with 53.3 $MtCO_2e$, 19.6 $MtCO_2e$, and 22.1 $MtCO_2e$ of reductions coming from the household, government, and investment sectors respectively. They contributed to −55.5%, −31.2%, −11.5% and −12.9% of the overall change in $N_2O$ emissions, respectively. Intermediate production technology lowered the $N_2O$ emissions by 17.7 $MtCO_2e$, making a percentage of change of −10.4%. In this respect, reductions in the household, government and investment sectors reached 14.7 $MtCO_2e$, 0.8 $MtCO_2e$, and 2.2 $MtCO_2e$ and accounted for −8.6%, −0.5% and −1.3% reductions, respectively. Driven by the total final consumption demand, the $N_2O$ emissions rose by 318.0 Mt $MtCO_2e$, of which 189.4 $MtCO_2e$, 32.1 $MtCO_2e$, and 96.5 $MtCO_2e$ came from household consumption, government consumption, and investment respectively. The respective percentages of change in the $N_2O$ emissions attained 186.0%, 110.8%, 18.7% and 56.5%. Due to the impact of final consumption demand structure, $N_2O$ emissions dropped by 34.4 Mt $MtCO_2e$ as a result of respective reductions of 21.9 $MtCO_2e$ and 12.6 $MtCO_2e$ in the household and investment sectors, and an increase of 0.09 $MtCO_2e$ in the government sector. The respective percentages of change in the $N_2O$ emissions stood at −20.1%, −12.8%, −7.3% and 0.1%, respectively.

Household final consumption demand is also the most influential factor for China's F-gases emissions. From 2004 to 2007, F-gases emissions were increased by the total final consumption demand, especially due to investment and intermediate production technology. Meanwhile, F-gases emissions were decreased due to the emissions intensity, noticeably in the investment sector. During this period, China's F-gases emissions increased by 26.0 $MtCO_2e$ overall. Affected by the emissions intensity, the F-gases emissions were reduced by 25.1 $MtCO_2e$, including 10.1 $MtCO_2e$, 4.7 $MtCO_2e$, and 10.3 $MtCO_2e$ in the household, government, and investment sectors respectively. They contributed to −96.5%, −38.9%, −18.3% and −39.4% of the overall change in F-gases emissions, respectively. Due to intermediate production technology, the F-gases emissions went up by 0.4 $MtCO_2e$, of which 0.08 $MtCO_2e$, 0.09 $MtCO_2e$, and 0.2 $MtCO_2e$ occurred in the household, government, and investment sectors, respectively. Therefore, the percentages of change in F-gases emissions registered 1.6%, 0.3%, 0.3% and 1.0%, respectively. Total final consumption demand drove the F-gases emissions up by 50.4 $MtCO_2e$, contributing to of 193.9% of the overall change in F-gases emissions. Incremental emissions related to household consumption, government consumption, and investment were 17.1 $MtCO_2e$, 7.4 $MtCO_2e$, and 25.9 $MtCO_2e$ and contributed to 65.9%, 28.5% and 99.5% of the overall change in F-gases emissions, respectively. The final consumption demand structure led to an increase of 0.3 $MtCO_2e$ in F-gases emissions, including 0.2 $MtCO_2e$, 0.007 $MtCO_2e$, and 0.05 $MtCO_2e$ in the household, government, and investment sectors, respectively. Their rates of contribution to the overall change in F-gases emissions registered 0.9%, 0.8%, 0.03% and 0.2%, respectively.

From 2007 to 2011, total final consumption demand, especially in investment, exerted an upward influence on F-gases emissions, while the emissions intensity, especially for the investment sector, and intermediate production technology affected the emissions in the opposite direction. During this period, China's F-gases emissions grew by 59.0 $MtCO_2e$ overall. Due to emissions intensity, the F-gases emissions were cut by 37 $MtCO_2e$, of which 13.2 $MtCO_2e$, 5.8 $MtCO_2e$, and 18.7 $MtCO_2e$ occurred in the household, government, and investment sectors respectively. The rate of contribution to the overall increase in F-gases emissions were calculated to be −60.4%, −22.4%, −9.9% and −31.7%, respectively. Intermediate production technology caused a decline of 3.7 $MtCO_2e$, contributing to −6.2% of the overall change in F-gases emissions. Specifically, 1.1 $MtCO_2e$, 0.2 $MtCO_2e$, and 2.4 $MtCO_2e$ were reduced in the household, government, and investment sectors respectively, representing −1.8%, −0.4% and −4.0%. Driven by total final consumption demand, F-gases emissions increased 106.8 $MtCO_2e$, of which 32.3 $MtCO_2e$, 11.6 $MtCO_2e$, and 62.9 $MtCO_2e$ were related to household consumption, government consumption, and investment, respectively. Their rates of contribution to the overall change in F-gases emissions registered 181.3%, 54.8%, 19.6%, and 106.8%. The final consumption demand structure contributed to −11.0% of the F-gases emissions by reducing 6.5 $MtCO_2e$. Specifically,

emissions related to household consumption and investment decreased by 1.9 MtCO$_2$e and 4.6 MtCO$_2$e respectively and emissions related to government consumption increased by 0.03 MtCO$_2$e, so respective percentages of change in F-gases emissions were −3.3%, 0.1% and −7.8%.

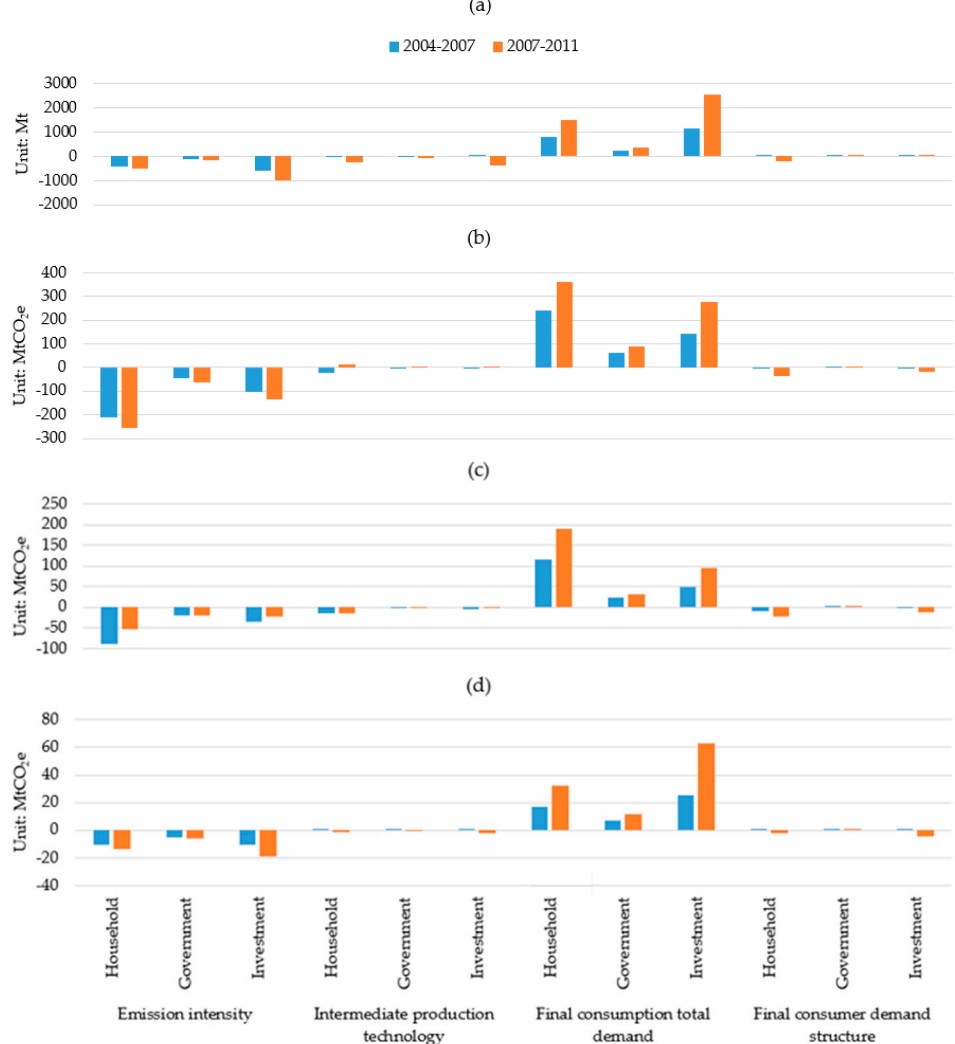

**Figure 3.** (**a**) China's indirect consumption-based CO$_2$ emissions by SDA; (**b**) China's indirect consumption-based CH$_4$ emissions by SDA; (**c**) China's indirect consumption-based N$_2$O emissions by SDA; (**d**) China's indirect consumption-based F-gases emissions by SDA.

*4.4. Discussion*

China has been the world's largest production-based emitter of GHGs since 2004. From the perspective of growth rate, CO$_2$ emissions far outpaced non-CO$_2$ GHG emissions, as the former's proportion in the overall emissions increased rapidly while the latter's proportion decreased. Among non-CO$_2$ GHG emissions, F-gases and N$_2$O emissions experienced the fastest growth in spite of their low baseline amount, which should not be ignored, considering the strong greenhouse effects of these two gases. At the same time, China's consumption-based emissions have accelerated, which has exceeded the growth rate of production-based emissions, resulting in a decline in net export emissions. The period of 2004–2011 is divided into two periods: 2004–2007 and 2007–2011. During these two periods, CO$_2$ production-based emissions and consumption-based emissions are increasing, but the growth rate of production-based emissions is decreasing. The growth rate of consumption-based emissions is rising; for non-CO$_2$ GHGs, both production-based and consumption-based emissions are accelerating. Whether it is consumption-based emissions or production-based emissions, China's GHG

emissions continue to grow, the gap in emissions is expanding year by year, and consumption-based emissions are always far below production emissions. In addition, China's per capita GHG emissions and consumption-based emissions have now exceeded global per capita GHG emissions. China's per capita GHG production emissions exceeded global per capita GHG emissions in 2007, while China's per capita GHG consumption-based emissions exceeded global per capita GHG emissions in 2011.

Our research has limitations in following aspects. Firstly, since the GTAP database is currently only updated to the year 2011, and the latest global emissions database is also only available for years before 2012, our research has only focused on the three years of 2004, 2007 and 2011, and does not reflect the latest changes in consumer emissions. Secondly, neither the emissions inventory data nor the GTAP database provide an uncertainty analysis, thus we couldn't estimate the uncertainty of the accounting results for consumption-based emissions. Last but not least, we may introduce further uncertainty in the process of mapping non-$CO_2$ emissions to economic sectors.

## 5. Conclusions

China is the biggest emitter and the largest net exporter of GHG emissions in the world, whether $CO_2$ or non-$CO_2$ GHGs. At present, China's per capita GHG emissions have exceeded the global average from both consumption and production perspectives. Our paper contributes to existing literature in two ways. First, for the first time we estimated China's consumption based GHGs using the latest available trade database (GTAP 9), the Second National Communication on Climate Change of the People's Republic of China and the People's Republic of China First Biennial Update Report on Climate Change. Second, we analyzed the factors driving the growth of China's consumption-based emissions using the SDA method.

From 2004 to 2011, China's GHG emissions continued to grow, with production-based emissions contributing more, but at a lower growth compared to consumption-based emissions. In terms of gas types, $CO_2$ contributed the most to added consumption-based emissions, followed by F-gases from the non-$CO_2$ GHG category. In terms of growth rate, $CO_2$ emissions increased rapidly, and at a larger magnitude than non-$CO_2$ GHG emissions, as the former took up a large proportion and the latter a decreasing proportion. In addition, F-gases showed the fastest growth among non-$CO_2$ GHGs, followed by $N_2O$. Considering their strong greenhouse effect, the added emissions should not be ignored even though the total emissions are limited.

In recent years, China's net exports of emissions declined as the growth of consumption-based emissions outpaced that of production-based emissions. During the periods from 2004 to 2007 and from 2007 to 2011, both production-based and consumption-based $CO_2$ emissions were on the rise, but growth slowed from the production perspective and picked up from the consumption perspective. As for non-$CO_2$ GHGs, both production-based and consumption-based emissions grew at a faster rate. Amid continuous growth in production-based and consumption-based GHG emissions, the net exports of GHG emissions began to decrease after peaking in 2007, and the net exports of non-$CO_2$ GHG emissions dropped by 55.2% in 2011 from the 2007 level. Thus, consumption-based emissions stayed much lower than production-based emissions, and the gap widens year by year. The SDA results show that investment as a portion of the total final consumption demand most affects consumption-based $CO_2$ emissions, and household consumption most affects consumption-based $CH_4$, $N_2O$ and F-gases emissions.

According to the IPCC Special Report on the impacts of global warming of 1.5 °C above pre-industrial levels, achieving the global warming target under the Paris Agreement requires in-depth mitigation of non-$CO_2$ GHG emissions while controlling $CO_2$ emissions. Otherwise, non-$CO_2$ GHG emissions will alter the remaining carbon budget and significantly reduce the probability of limiting global warming to 1.5 °C. Non-$CO_2$ GHG emissions are mainly driven by household consumption, and the accurate accounting of consumption-based emissions will help formulate effective policies from the consumption side to substantially reduce non-$CO_2$ GHG emissions. Our analysis suggests that the growth of China's consumption-based emission is changing from an investment-driven pattern

towards a consumption-driven pattern. Thus, the reduction of non-$CO_2$ emissions requires a combined policy package between the up-stream mitigation measures and the change of consumer behavior in the demand side. Compared with the mitigation of $CO_2$, the change of consumer behavior is critical for the long-term reduction of non-$CO_2$ emissions.

**Author Contributions:** Conceptualization, F.T. and H.G.; methodology, F.T. and H.G.; software, H.G.; validation, F.T., A.G. and G.W.; formal analysis, H.G. and F.T.; writing—original draft preparation, H.G. and F.T.; writing—review and editing, H.D., F.T., A.G. and G.W.

**Funding:** This research was funded by the National Key Research and Development Program of China (No. 2016YFA0602702) and the National Natural Science Foundation of China (No. 71673162 and No. 71173131).

**Acknowledgments:** The authors also acknowledge Cecilia Han Springer for editing this manuscript.

**Conflicts of Interest:** The authors declare no conflict of interest.

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
