# Peer review of "A Structural Decomposition Analysis of China’s Consumption-Based Greenhouse Gas Emissions"

_energies, doi:10.3390/en12152843_

Round 1
Reviewer 1 Report
The authors try to examine China 'based on GHG emissions and their influencing factors, with a focus on the composition of these emissions, making a comparative analysis of production-based emissions, based on the emissions, and based on the emission intensity, and a quantitative analysis of China's indirectly-based emissions. However, the authors have had partial success in achieving their objectives. In fact, I have some major comments.
Major comments
1.- The main problem with the manuscript is that it is purely descriptive, without providing any evidence of significance, or any type of inference, even if it is simple.
The authors should make some inference, even if it is simple. For example, significance of the correlations, analysis of the trend (with confidence intervals), etc., etc.
2.- The discussion should be called results.
The authors should include a real discussion section, in which they explain and compare their findings with similar articles or with similar methodology, or on the same field. In addition, they should include, at least one paragraph, limitations.
Author Response
1.- The main problem with the manuscript is that it is purely descriptive, without providing any evidence of significance, or any type of inference, even if it is simple.
The authors should make some inference, even if it is simple. For example, significance of the correlations, analysis of the trend (with confidence intervals), etc., etc.
Response: Thank you for your helpful comments. Our study accounts for consumption-based emission by mapping existing GHGs inventory data with the input-output economic and trade tables. The given GHG inventories (collected from EPA, EDGAR and China BUR) and input-output tables (from GTAP) are lack of statics analysis including confidence level and significance, thus it is difficult for us to conduct an inference analysis on the significance and confidence level. However, we realized the importance of such study in the future if such data is available for inventories and IO tables.
To address the concerns from the reviewer, we have added a further explanation at the end of section 3.
“The accurate calculation of consumption-based emissions must be based on an accurate inventory of current emissions and input-output data. However, the accuracy of non-CO2 inventory is much less certain than that of CO2 inventory. For example, the accuracy of China’s non-CO2 inventories ranges from ±55-60% at the high end to ±nearly 15% at the low end, compared with an uncertainty range of ±6% for China’s CO2 emissions. Due to the lack of statistical data, it is currently impossible to quantify the uncertainty range of accounting results of consumption-based emissions. Improving the accuracy of inventory data can greatly help improve the accuracy of consumption-based emissions accounting.”
2.- The discussion should be called results.
The authors should include a real discussion section, in which they explain and compare their findings with similar articles or with similar methodology, or on the same field. In addition, they should include, at least one paragraph, limitations.
Response: Thank you for your helpful comments. To address your comments, we added a summary section in 4.4 including both discussions on main results and limitations. We also added a comparison of our result with existing literature on page 8.
Reviewer 2 Report
The authors have to relate their analysis to Global Warming Potentials of the pollutants considered. This is important in such analyses.
The data are outdated referring to 2004 - 2011.
Similarly the references are outdated.
Author Response
Comment: The authors have to relate their analysis to Global Warming Potentials of the pollutants considered. This is important in such analyses.
Response: Thank you for your helpful comments. To address your comments, we explained the GWPs we used on page 2. To be consistent with existing inventory reporting guidelines, we use the GWPs from the 1996 reporting guidelines of the IPCC, the reporting guideline used by China to UNFCCC.
Comment: The data are outdated referring to 2004 - 2011.
Response: Thank you for your helpful comments. The outdated data is largely limited by the available data of non-CO2 and IO. The latest non-CO2 inventory is only available for years before 2012, and for IO tables, the latest GTAP IO tables are for 2011. We added a paragraph on the limitation of this paper at the end of sector 4 to address your comments.
Our research has limitations in the following aspects: Firstly, since the GTAP database is currently only updated to the year 2011, and the latest global emissions database is also only available for years before 2012, our research has only focused on the three years of 2004, 2007 and 2011, and does not reflect the latest changes in consumer emissions. Secondly, neither the emissions inventory data nor the GTAP database provides an uncertainty analysis, thus we cannot estimate the uncertainty of the accounting results for consumption-based emissions. Last but not least, we may introduce further uncertainty in the process of mapping non-CO2 emissions to the economic sectors.
Comment: Similarly the references are outdated.
Response: Thank you for your helpful comments. To address your comments, we added more than a dozen recent related references in different sections of the paper.
Reviewer 3 Report
This paper presents an analysis of CO2 and non-CO2 GHG emissions in China with a structural decomposition analysis of the results. The major issues I have with the paper are:
Little to no information is given about how the inventory for non-CO2 GHGs was built. This statement is made on page 5: "Non-CO2 GHG emissions are from the database built in this paper." However, where were the activity factors and emissions factors for each of the gases and emissions categories/sectors obtained?
The whole discussion section seems like the information could be given in a table or graphically. Results from the analyses are given in words as they are shown in the figures and tables but are not sufficiently described in context; how do they compare with others’ estimates? What new information does this paper give about mitigation strategies that we did not have before? The lack of interpretation of what the results mean make it difficult to determine the impact of the paper.
Other specific comments: The years of analysis (2004, 2007, and 2011) should be given in the abstract. Why is 2011 the most recent year of data? Isn’t there more recent inventory data to use? Page 2 lines 45-50: Delete "on the one hand" and "on the other hand." Should read: "Non-CO2 GHG emissions represent a smaller portion … modeling analysis. Additionally, the amount of research that has been done is less substantial and thus does not draw as much attention to non-CO2 GHG emissions." Replace "is still great space…" with "are opportunities to reduce non-CO2 GHG emissions that would have a significant impact on overall GHG emissions reductions." Page 2 lines 60-63 are unnecessary. The section headings are sufficient. First sentence of section 2. Literature review needs a reference. Page 2 lines 50-52: This sentence has a contradictory statement further down, lines 67-68. Be clear about how this paper is new or different from previous studies of China’s consumption-based GHG emissions. Page 5 lines 211-228: This section should be reworded. There are contradictions; the first couple sentences state that annual consumption-based GHG emissions in China were 4.0 GtCO2e in 2011. In line 218 it says that China’s consumption-based GHG emissions were 8.7 GtCO2e in 2011. Which is correct? There are also repetitive statements. Overall this paragraph could use some improved writing so the message is clear, concise, and most importantly, correct. Figure 2 would benefit from labels of CH4, N2O and F-gases above (b), (c), and (d). This could also simplify the caption. ("Composition of China’s consumption-based CH4 (b), N2O (c), and F-gases (c) in 2004, 2007, and 2011.") Figure 3 is placed too far down in the text. It is first referred to on page 8 but appears on page 11.
Page 12 lines 505-506: I don’t know that I agree with this statement: "Non-CO2 GHG emissions are relatively dispersed and highly uncertain, so they can be more difficult to control from the production side." Can you provide references?
Author Response
Comment: Little to no information is given about how the inventory for non-CO2 GHGs was built. This statement is made on page 5: "Non-CO2 GHG emissions are from the database built in this paper." However, where were the activity factors and emissions factors for each of the gases and emissions categories/sectors obtained?
Response: Thank you for this suggestion, we added detailed information on how we built non-CO2 GHG inventory in lines 235-255 on page 6.
The data sources of non-CO2 GHG are from the Second National Communication on Climate Change of the People’s Republic of China[1] and the People’s Republic of China First Biennial Update Report on Climate Change[2]. According to relevant decisions of United Nations Framework Convention on Climate Change (UNFCCC), and considering China’s circumstances, the National Greenhouse Gas Inventory of 2005 and 2012 covers six gases including carbon dioxide (CO2), methane (CH4), Nitrous oxide (N2O), hydrofluorocarbons (HFCs), perfluorocarbons (PFCs) and sulfur hexafluoride (SF6) from energy, industrial processes, agriculture, land use change and forestry and waste. The Inventory mainly follow s the Revised 1996 IPCC Guidelines for National Greenhouse Gas Inventories (hereinafter referred to as the Revised 1996 IPCC Guidelines) and the IPCC Good Practice Guidance and Uncertainty Management in National Greenhouse Gas Inventories (hereinafter referred to as the IPCC Good Practice Guidance). Activity data are mainly from official statistics, while emission factors are mainly from the 2012 China’s country-specific parameters5. In addition, this paper uses the IPCC definition code for non-CO2 greenhouse gas activities and China's GDP growth rate of World Bank Open Data to adjust non-CO2 GHG activities and emissions to 2004, 2007 and 2011 reference years as well as 57 GTAP sectors.
Comment: The whole discussion section seems like the information could be given in a table or graphically. Results from the analyses are given in words as they are shown in the figures and tables but are not sufficiently described in context; how do they compare with others’ estimates? What new information does this paper give about mitigation strategies that we did not have before? The lack of interpretation of what the results mean makes it difficult to determine the impact of the paper.
Response: Thank you for your comment. First of all, we added a table on page 8 to summarize the discussion section. Secondly, we added the comparison between this paper and the previous study in lines 355-360 of page 9. Finally, we summarize the policy implication of this study at the end of the conclusion section.
Table 1. China's GHG emissions in 2004, 2007 and 2011
MtCO2e | Production-based emissions | Consumption-based emissions | Direct household | Indirect household | Investment | Direct government | Indirect government | |
2004 | CO2 | 4723.6 | 3582.3 | 358.1 | 1310.4 | 1606.1 | 0.0 | 307.6 |
CH4 | 905.5 | 749.5 | 0.3 | 440.3 | 216.8 | 0.0 | 92.3 | |
N2O | 385.6 | 322.6 | 1.0 | 211.7 | 74.8 | 0.0 | 36.2 | |
F-gases | 130.2 | 74.4 | 0.0 | 28.6 | 35.1 | 0.0 | 10.7 | |
GHG | 6144.9 | 4728.8 | 359.4 | 1991.1 | 1932.8 | 0.0 | 446.8 | |
2007 | CO2 | 6468.3 | 4853.1 | 441.1 | 1790.5 | 2196.5 | 0.0 | 425.0 |
CH4 | 954.9 | 803.5 | 0.5 | 447.1 | 247.4 | 0.0 | 109.0 | |
N2O | 387.6 | 335.5 | 1.2 | 215.0 | 81.9 | 0.0 | 38.5 | |
F-gases | 187.3 | 100.3 | 0.0 | 35.9 | 51.0 | 0.0 | 13.4 | |
GHG | 7998.0 | 6092.5 | 442.8 | 2488.6 | 2576.9 | 0.0 | 585.9 | |
2011 | CO2 | 8465.6 | 6969.5 | 590.9 | 2364.7 | 3466.4 | 0.0 | 547.5 |
CH4 | 1112.0 | 1044.2 | 1.9 | 530.5 | 375.9 | 0.0 | 137.8 | |
N2O | 558.3 | 506.5 | 1.5 | 314.5 | 141.8 | 0.0 | 50.2 | |
F-gases | 172.3 | 159.2 | 0.0 | 52.0 | 88.3 | 0.0 | 18.9 | |
GHG | 10308.3 | 8679.4 | 594.3 | 3261.7 | 4072.3 | 0.0 | 754.5 |
Our analysis suggests that the growth of China’s consumption-based emission is changing from an investment-driven pattern towards a consumption-driven pattern. Thus, the reduction of non-CO2 emissions requires a combined policy package between the up-stream mitigation measures and the change of consumer behavior in the demand side. Compared with the mitigation of CO2, the change of consumer behavior is critical for the long-term reduction of non-CO2 emissions.
Other Specific Comments:
Comment: The years of analysis (2004, 2007, and 2011) should be given in the abstract.
Response: Thank you for your helpful comments. We added those years in the abstracts.
Comment: Why is 2011 the most recent year of data? Isn’t there more recent inventory data to use?
Response: Thank you for your helpful comments. The outdated data is largely limited by the available data of non-CO2 and IO. The latest non-CO2 inventory is only available for years before 2012, and for IO tables, the latest GTAP IO tables are for 2011. We added a paragraph on the limitation of this paper at the end of sector 4 to address your comments.
Comment: Page 2 lines 45-50: Delete "on the one hand" and "on the other hand." Should read: "Non-CO2 GHG emissions represent a smaller portion … modeling analysis. Additionally, the amount of research that has been done is less substantial and thus does not draw as much attention to non-CO2 GHG emissions." Replace "is still great space…" with "are opportunities to reduce non-CO2 GHG emissions that would have a significant impact on overall GHG emissions reductions."
Response: Thank you, we have accepted your helpful advice and modified these paragraphs in lines 47-51 and lines 55-56 of page 2.
Comment: Page 2 lines 60-63 are unnecessary. The section headings are sufficient.
Response: Thank you for reminding us, we have deleted lines 64-66 of page 2.
Comment: First sentence of section 2. Literature review needs a reference.
Response: Thanks a lot, we added the citation for the first sentence of section 2 on page 2.
Comment: Page 2 lines 50-52: This sentence has a contradictory statement further down, lines 67-68. Be clear about how this paper is new or different from previous studies of China’s consumption-based GHG emissions.
Response: Many thanks, to address the comments, we added a para at the end of para3 on page 2 to explain how our paper differs from existing literature.
In summary, this study differs from existing literatures:First, the existing research only focuses on the accounting of consumption-based emissions for specific individual greenhouse gas (CO2 or CH4), while this study is the first to account for the consumption-based emissions of all greenhouse gases in China. Secondly, existing studies have not analyzed the main drivers driving changes in China's consumption-based emissions, while this paper uses the SDA method to analyze this in detail.
Comment: Page 5 lines 211-228: This section should be reworded. There are contradictions; the first couple of sentences state that annual consumption-based GHG emissions in China were 4.0 GtCO2e in 2011. In line 218 it says that China’s consumption-based GHG emissions were 8.7 GtCO2e in 2011. Which is correct? There are also repetitive statements. Overall this paragraph could use some improved writing so the message is clear, concise, and most importantly, correct.
Response: Many thanks, we have revised it in lines 265-267 on page 6.
From 2004 to 2011, both production-based and consumption-based GHG emissions in China were on the rise, with increments of 4.2 GtCO2e and 4.0 GtCO2e (from 4.7 GtCO2e to 8.7 GtCO2e during 2004–2011) respectively, or with an increase of 67.8% and 83.6% respectively.
Comment: Figure 2 would benefit from labels of CH4, N2O and F-gases above (b), (c), and (d). This could also simplify the caption. ("Composition of China’s consumption-based CH4 (b), N2O (c), and F-gases (c) in 2004, 2007, and 2011.")
Response: Thanks, we simplified the labels of figure 2 on page 10.
Comment: Figure 3 is placed too far down in the text. It is first referred to on page 8 but appears on page 11.
Response: Thank you, we moved figure 3 to page 11.
Comment: Page 12 lines 505-506: I don’t know that I agree with this statement: "Non-CO2 GHG emissions are relatively dispersed and highly uncertain, so they can be more difficult to control from the production side." Can you provide references?
Response: Thank you for reminding us, we have removed this imprecise statement in lines 608-610 of page 15.
[1] http://data.ncsc.org.cn/portals/word-news-detail.html?column=statistical-report&&id=148744, 2019.6.29
[2], 5 https://unfccc.int/sites/default/files/resource/chnbur1.pdf, 2019.6.29
Round 2
Reviewer 1 Report
The authors have answered all my comments and have incorporated most of them in the new version of the manuscript. I have no further comments.
Reviewer 2 Report
Satisfied with the way tthe authors have tackled my comments
Reviewer 3 Report
I have reviewed the author's responses to my comments as well as the revised manuscript. All were addressed sufficiently. I recommend publication.